# Do Multiple Sex/Gender Dimensions Play a Role in the Association of Green Space and Self-Rated Health? Model-Based Recursive Partitioning Results from the KORA INGER Study

**DOI:** 10.3390/ijerph20075241

**Published:** 2023-03-23

**Authors:** Lisa Dandolo, Klaus Telkmann, Christina Hartig, Sophie Horstmann, Sara Pedron, Lars Schwettmann, Peter Selsam, Alexandra Schneider, Gabriele Bolte

**Affiliations:** 1Department of Social Epidemiology, Institute of Public Health and Nursing Research, University of Bremen, 28359 Bremen, Germany; 2Health Sciences Bremen, University of Bremen, 28359 Bremen, Germany; 3Institute of Health Economics and Health Care Management, Helmholtz Zentrum München, German Research Center for Environmental Health, 85764 Neuherberg, Germany; 4Professorship of Public Health and Prevention, Technical University of Munich, 80992 Munich, Germany; 5Division for Health Economics, Department of Health Services Research, School of Medicine and Health Sciences, Carl von Ossietzky University of Oldenburg, 26129 Oldenburg, Germany; 6Department Monitoring and Exploration Technologies, Helmholtz Centre for Environmental Research—UFZ, 04318 Leipzig, Germany; 7Institute of Epidemiology, Helmholtz Zentrum München, German Research Center for Environmental Health, 85764 Neuherberg, Germany

**Keywords:** sex, gender, intersectionality, model-based recursive partitioning, decision trees, green spaces

## Abstract

Exposure to green space has a positive impact on health. Whether sex/gender modifies the green space–health association has so far only been studied through the use of a binary sex/gender category; however, sex/gender should be considered more comprehensively as a multidimensional concept based on theoretical approaches. We therefore explored whether sex/gender, operationalized through multiple sex/gender- and intersectionality-related covariates, modifies the green space–self-rated health association. We collected data from participants involved in the German KORA study (Cooperative Health Research in the Region of Augsburg) in 2019. Self-rated health was assessed as a one-question item. The availability of green spaces was measured subjectively as well as objectively. The multiple sex/gender- and intersectionality-related covariates were measured via self-assessment. To analyze the data, we used model-based recursive partitioning, a decision tree method that can handle complex data, considering both multiple covariates and their possible interactions. We showed that none of the covariates operationalizing an individual sex/gender self-concept led to subgroups with heterogeneous effects in the model-based tree analyses; however, we found effect heterogeneity based on covariates representing structural aspects from an intersectionality perspective, although they did not show the intersectional structuring of sex/gender dimensions. In one identified subgroup, those with a lower education level or a feeling of discrimination based on social position showed a positive green space–self-rated health association, while participants with a higher education level or no feeling of discrimination based on social position had a high level of self-rated health regardless of the availability of green spaces. Model-based recursive partitioning has the potential to detect subgroups exhibiting different exposure–outcome associations, with the possibility of integrating multiple sex/gender- and intersectionality-related covariates as potential effect modifiers. A comprehensive assessment of the relevance of sex/gender showed effect heterogeneity based on covariates representing structural aspects from an intersectionality perspective.

## 1. Introduction

Exposure to green space is considered to have a positive impact on both physical and mental health [1,2,3,4]. Many pathways for this green space–health impact have been suggested. Markevych [2] structured them in three categories: (1) green space reduces harm through reductions in noise, air pollution, and heat exposure; (2) green space restores personal capacities by reducing stress and restoring attention; and (3) green space builds new capacities by encouraging physical activity and social cohesion [2,3,5]. However, the effect of green space on health is not consistent across all population groups. For instance, stronger green space–health associations were found for people living in more urban areas compared to rural areas [6]. Interestingly, people with a lower socioeconomic status (SES) showed a more beneficial green space effect than more affluent people, especially in Europe [7]. At the same time, people with a lower SES still have less access to green space [8,9]; thus, increasing the access to green space for disadvantaged people can increase health equity [1,7].

One important social determinant of health is sex/gender [10]. Whether the green space–health association is modified by sex/gender has so far only been investigated through the use of a binary sex/gender category [11]. In a recent review it was suggested that women show a stronger positive association of green space and health compared to men, mainly for obesity-related health outcomes and mortality [12]; however, for other health outcomes the results were inconsistent [11,12]. For instance, for self-rated health some studies have shown that the association with green space was stronger for women [13], while in others it was stronger for men [14], and in some cases no differences between the two groups were found [15,16]. As possible reasons for differences between women and men, researchers discussed, among others, park safety issues [17], differences in physical activity frequency and type [18], time spent in green space due to care activities [16,19], or differences in the psychological relationships with green space [12]. The heterogeneity of the results concerning a possible effect modification by sex/gender of the green space–health association may be partly explained by the fact that both the green space exposures and health outcomes are measured inconsistently. Most importantly, however, none of the reviewed studies considered the complexity of sex/gender adequately [11,12]. Sex/gender was only used as a binary category, the terms sex and/or gender were not clearly defined and often used interchangeably, leading to a conceptual muddle, and most studies gave no definite description as to how the binary sex/gender data were obtained [11].

Within the INGER (integrating gender into environmental health research) project, we define sex/gender as a non-binary, structural category that has multiple biological (sex) and social (gender) dimensions [20]. We use the expression “sex/gender” to indicate that sex and gender are entangled, as conceptualized in the embodiment theory [20,21]. We developed a multidimensional sex/gender concept, describing a sex/gender self-concept at the individual level, which arises within the context of social and cultural sex/gender relations. Furthermore, we conceptualize sex/gender as interacting with other intersecting categories of social inequality and power relations [20].

For years now, researchers have been calling for a more comprehensive integration of sex/gender into health research [10,22,23,24]; however, across different fields of environmental health research sex/gender is still considered solely as a binary construct without referring to any sex/gender theoretical concepts [11,25]. The aim of this study was therefore to explore whether sex/gender, considered as a multidimensional structural concept, modifies the association of green space and self-rated health.

We operationalized the multidimensional sex/gender concept [20] through multiple sex/gender- and intersectionality-related variables [26]. This required us to use statistical methods that can address a high number of covariates. As we looked at sex/gender from an intersectionality perspective, it was equally important to find methods that not only consider multiple covariates but also interactions between them. In a previous paper we identified decision trees as adequate methods with which to include multiple sex/gender- and intersectionality-related variables as well as identify interactions [27]; however, common decision tree methods, such as classification and regression trees (CARTs; Breiman [28]), as well as conditional inference trees (CITs; Hothorn [29]), are designed to find subgroups with differences in a dependent variable, e.g., a health outcome [30] or environmental exposure [27]. Thus, these methods can only capture descriptive intersectional research, as described by Bauer and Scheim [31]. In the present study, however, we wanted to apply an analytic intersectional approach [31] and test effect heterogeneity, in this manner finding subgroups characterized by multiple sex/gender- and intersectionality-related variables with differential effects of an environmental exposure on a health outcome. This cannot be achieved by using conventional decision trees. We therefore applied model-based trees as they find subgroups based on fitting generalized linear models [32].

## 2. Materials and Methods

Appendix A shows a research structure diagram with which to visually summarize the materials and methods presented in the following sections as well as provide a concise overview of the research process of our main analyses.

### 2.1. INGER Project

As described in Dandolo [27], the collaborative research project INGER (https://www.uni-bremen.de/en/inger, accessed on 25 May 2022) focused on establishing innovative sex/gender-sensitive methods for data collection and analyses in population-based environmental health studies. To improve sex/gender data collection, the INGER research team introduced a multidimensional sex/gender concept [20] and constructed new sex/gender questionnaire modules [26], which were tested within the KORA (Cooperative Health Research in the Augsburg Region) platform [33].

### 2.2. Study Population

As previously described [26,27,33], the KORA platform was designed to investigate the relationships between health, diseases, and living conditions in the population of Augsburg and two adjacent districts in Southern Germany. In brief, four cross-sectional surveys (S1–S4) at five-year intervals and various follow-ups have been implemented since 1984: S1 in 1984/85 (participants born between 1920 and 1959), S2 in 1989/90 (1915–1964), S3 in 1994/95 (1920–1969), and S4 in 1999–2000 (1925–1975). The paper-based INGER questionnaire, with the newly constructed sex/gender questionnaire modules [26] and a comprehensive set of questions on residential green spaces, was sent by letter to KORA participants aged 44–93 in 2019 (N = 5256). These comprised all of the participants of the KORA FIT study, which was distributed shortly before the INGER questionnaire in 2018/2019. The KORA FIT study included participants from all four surveys with a current age of 53–74 years. The INGER questionnaire was additionally sent to the 49–53 year olds from S3 and to the 49–53 year olds as well as the 74–93 year olds from S4. More information about the INGER KORA population can be found in Kraus [26].

### 2.3. Sex/Gender- and Intersectionality-Related Covariates

We chose the same 40 sex/gender- and intersectionality-related covariates, based on our multidimensional INGER sex/gender concept [20], that we used and described in more detail in our previous paper [27]. In brief, we used 17 covariates to operationalize the individual sex/gender self-concept (one covariate to operationalize *sex assigned at birth;* one for *current sex/gender identity*; twelve for *internalized sex/gender roles*; and three for *externalized sex/gender expressions)* and a total of 23 covariates to represent the structural sex/gender relationships (nine covariates to operationalize the *experience of discrimination*; eight for *care and household activities*; and six represent *intersectionality-related social categories* (e.g., education, employment, and income)). Appendix A provides a complete overview of the 40 sex/gender- and intersectionality-related covariates, including the questions and possible response categories.

### 2.4. Outcome

We used self-rated health as a valid outcome measure for both mental and physical health [34]. In the one-question item participants were asked how their health was in general. The question had five possible response categories (very good, good, moderate, bad, and very bad). As some of the categories had very small sample sizes, we combined the first two categories—very good (N = 269) and good (N = 1657)—into the category “good” and the remaining three categories—moderate (N = 550), bad (N = 57), and very bad (N = 1)—into the category “bad”. Dichotomizing the outcome variable also makes interpreting results from the model-based tree analyses more intuitive, as it allows us to apply a binary logistic regression model.

### 2.5. Exposure Variables

As suggested by Labib [35], we assessed and operationalized green space exposure in multiple ways in order to achieve a comprehensive evaluation of the green space–health association. We used both objective normalized difference vegetation index (NDVI) data and subjectively perceived green space measures, since it has been shown that they do not always coincide [36]. As has been frequently suggested, we also considered the self-rated quality of public green space rather than only its availability [2,37,38]. In total, we conducted analyses for four different exposure variables of green space. For the subjective, i.e., self-rated, measurements we used items from our KORA INGER questionnaire, while objective measurements were based on NDVI data accessible for the participants’ residential addresses.

#### 2.5.1. Access to High-Quality Public Green Spaces (Subjectively Measured)

The self-rated exposure variable *access to high-quality public green spaces* was constructed by using the information from three KORA INGER questions. In the first question, participants were asked if they had access to public green spaces in their residential environments. Parks, forests, or meadows were listed as examples, and the response categories were yes and no. Participants with access to public green spaces were subsequently split based on two follow-up questions. In these follow-up questions participants expressed (on a five-point scale from “strongly agree” to “strongly disagree”) whether they felt that the public green spaces in their residential environments were of high quality and whether they were well-maintained. Using a dichotomized exposure variable makes interpreting results from the model-based tree analyses more intuitive. Therefore, if participants responded with “strongly agree” to both follow-up questions they were placed in the category *high-quality public green spaces*. The remaining participants, together with the participants who stated that they did not have access to public green spaces, were placed in the response category *lower quality or no public green spaces*.

#### 2.5.2. Greenness in the Residential Environment (Subjectively Measured)

The self-rated exposure variable *greenness in the residential environment* was based on a one-question item in which the participants could rate the greenness of their neighborhoods by using four different response categories (very green, a little green, hardly green, and not green at all). For this assessment they were asked to consider all kinds of green spaces, i.e., green strips along streets as well as gardens and parks. As using a dichotomized exposure variable makes interpreting results from the model-based tree analyses more intuitive, we grouped the last three categories (a little green (N = 477), hardly green (N = 44), and not green at all (N = 5)) together into the new category “less green”.

#### 2.5.3. Greenness within a 300 or 1000 m Buffer around the Residential Address (Objectively Measured)

As was the case in our previous paper [27], we used NDVI data from the year 2019 for the two objectively measured exposure variables to determine vegetation density, i.e., greenness [39]. As previously described [27], we used Landsat 8 Operational Land Imager (OLI) satellite imagery with a resolution of 30 m in addition to Sentinel-2 (S2) imagery with a ground resolution of 10 m and cloud cover of less than 1% for single images. The left panel of Figure 1 shows the S2 imagery for the KORA study area—the city of Augsburg and two adjacent districts in Southern Germany. We used Google’s Earth Engine Code Editor (https://code.earthengine.google.com, accessed on 4 October 2021) for all calculations and image as well as quality selection. The included images were collected between March and October and calculated as the median of all accepted images. Here, the atmospherically corrected reflectance of near-infrared (NIR) and visible red light (RED) was used according to the following standard formula: NDVI = (NIR − RED) / (NIR + RED). The possible range of NDVI values is from −1 to +1. Values around 0 and lower either indicate densely populated areas or snow, water, rock, or sand. In comparison, values close to +1 indicate photosynthetically active plants in a high density [40,41]. Water areas were masked according to a Copernicus Global Land Cover Map [42], and negative values were set to missing values (NAs). NDVI data were obtained for each participant within a specified buffer around their residential address. As demonstrated for an exemplary address in the right panel of Figure 1, we chose a 300 m buffer for an area within easy walking distance and a 1000 m buffer for a more extended area for direct comparison.

### 2.6. Model-Based Recursive Partitioning

Subgroups were identified by a decision tree algorithm [27,28]. Since we were particularly interested in exposure–outcome relationships, classical decision tree methods, such as CARTs and CITs, were not applicable; however, model-based recursive partitioning [32] served this purpose. In particular, this algorithm fits a generalized linear model to the data to estimate the association between exposure and outcomes. As we dichotomized our outcomes, we could apply a binary logistic regression model. This first step led to estimates of the intercept and an exposure-specific coefficient. In a second step, a parameter instability test was employed to determine a splitting covariate. In principle, the algorithm checks whether a potential splitting covariate leads to substantial changes in the regression coefficients. If the null hypothesis of no parameter instability with respect to any of the potential splitting candidates could not be rejected at a significance level of at least α = 0.05, no split was induced. Thus, this test served as a stopping criterion. For technical details we referred to Zeileis and Hornik [43]. If a splitting variable was identified, a binary partition of the data was searched for by determining a dichotomization of the splitting variable, such that the negative log-likelihood was minimized in both resulting subnodes. This procedure was repeated at any emerging subnode until no parameter instability was detected anymore. We employed two additional stopping criteria: First, if a split would produce at least one node containing less than 100 observations, no split was induced. Second, the maximum depth of the tree was set to three, i.e., subgroups could not be defined by more than three consecutive interactions. These criteria preserved the interpretability of the results.

The resulting subgroups differed with respect to their regression coefficients. Note that effect heterogeneity depends on the definition of the conditional average treatment effect (CATE). We focus on odds ratios since these are applicable to binary as well as continuous exposures. As a sensitivity analysis for binary exposures, we also calculated the risk differences, $P\left[Y\right|E=1]-P\left[Y\right|E=0]$, for each node, i.e., the differences in outcome probabilities between the exposed and non-exposed. Confidence intervals for these quantities were calculated based on 2000 bootstrap samples. Recall that this measure depends on both regression coefficients. To our knowledge there exists no decision tree framework applicable to both continuous and binary exposures that explicitly focusses on differences in odds ratios. In principle, subgroups may exhibit identical odds ratios but differ substantially in intercepts, i.e., the prevalence of good self-rated health. To interpret the differences between the regression coefficients of two subgroups as effect heterogeneity, it was therefore necessary to determine whether the odds ratios differed between these two subgroups or only the prevalence of good self-rated health. This was carried out by comparing parameter estimates and confidence intervals in the final subgroups. Confidence intervals were calculated based on robust standard errors, since a weighting procedure was applied as described in the next subsection.

### 2.7. Inverse Probability Weighting

Exposures cannot be assumed to be randomized, since we were analyzing an observational study. In order to adjust for confounding variables and create a pseudo-randomized sample, we employed inverse probability weighting (IPW) based on propensity scores [44,45]. The propensity score, $P\left(E_{i}=1\right|X_{i})$, for observation i is defined as the conditional probability of being exposed given the confounding variables, $X_{i}$. For all four exposure measures we estimated the propensity scores by using the following confounders: age, house ownership status, degree of urbanization, self-rated financial situation, school and vocational education, employment status, family situation, and discrimination experiences based on ethnicity.

For the binary exposure variables propensity scores were estimated via logistic regression. Inverse probability weights were then defined as the reciprocal of the propensity score for exposed subjects and as the reciprocal of 1 minus the propensity score for non-exposed subjects. Since these weights can become quite large, we used a stabilized version for the subsequent analyses [46]. Thus, the weights for each observation were multiplied by the marginal probability of their observed exposure status. For the continuous NDVI exposure measures this approach was not feasible; however, an extension of this concept, called generalized propensity scores, was applicable [47]. For instance, in a non-discrete setting the conditional probability had to be replaced by the conditional probability density, $f\left(E_{i}\right|X_{i})$. This quantity was estimated via kernel density estimation. Weights were then defined similarly for each observation as the reciprocal of its conditional probability density.

To assess the balance of the weighted dataset, several balance diagnostics have been proposed [48]. In cases of a binary exposure, we compared the absolute standardized mean differences before and after adjustment. For a continuous confounding covariate, this quantity measures the standardized difference in means between the exposed and non-exposed. In cases of categorical confounders, the standardized difference in proportions was measured. According to Austin [49], a dataset can be considered balanced if the absolute standardized mean difference is below 0.1 for each covariate. In contrast, for continuous exposures the absolute exposure–covariate correlations were investigated.

Moreover, in order to consistently estimate average exposure effects, the positivity or common support assumption needs to hold. In particular, there should not exist any combination of covariates such that the probability of being exposed or non-exposed is zero. For binary exposures this assumption can be checked visually by searching for a lack of overlap in the plots of the distributions of the propensity scores stratified by exposure status. Additionally, we investigated the mean and maximum of the stabilized inverse probability weights, since a mean being far from one or extreme values suggest a violation of the positivity assumption, as pointed out by Cole and Hernán [50].

### 2.8. Additional Analyses for the INGER KORA FIT Subsample with 53 Sex/Gender- and Intersectionality-Related Covariates

The KORA FIT survey contained additional information that could also be interpreted with respect to our multidimensional INGER sex/gender concept [20]. For this reason, we conducted some additional analyses with a total of 53 covariates (i.e., with the original 40 covariates plus 13 additional covariates from the KORA FIT survey) with all of the INGER participants that also took part in the KORA FIT study. Six of the thirteen new covariates were covariates contributing to operationalize *intersectionality-related social categories* (e.g., occupation, number of household members, and degree of disability), three covariates represented *health-related behaviors* (e.g., smoking behavior, alcohol consumption,) and four covariates represented *psychosocial factors* (e.g., perceived stress, self-efficacy). The questions and response categories for the additional 13 covariates are presented in Appendix A. For the propensity score estimation for these additional analyses we used the same confounders as in the main analyses, additionally adding the equivalent income variable and employment categories variable that were available here. Appendix A shows a research structure diagram with which to visually summarize and provide a concise overview of the research process of our additional analyses for the INGER KORA FIT subsample.

### 2.9. Additional Analysis with a Binary Sex/Gender Variable

Sex/gender has multiple biological and social dimensions [20], and we therefore operationalized it through multiple covariates in our main analyses; however, recent reviews [12,20] have shown that all former studies examining whether sex/gender modifies the association of green space and health solely used a binary sex/gender variable, comparing men to women. For a direct comparison with former research studies and in order to contrast them with our more advanced sex/gender analyses, we conducted additional analyses with only the binary sex/gender covariate *SexAtBirth*. For these analyses we used logistic regressions with a green space exposure x *SexAtBirth* interaction term weighted by inverse probability weights based on the propensity score.

### 2.10. Software

All analyses were conducted in R version 4.0.2. Inverse probability weights were estimated by using the *weightit* function in the *WeightIt* package for both binary and continuous exposures, as described above. The *cobalt* package provides additional balance diagnostics. Model-based recursive partitioning is implemented in the *glmtree* function contained in the *partykit* package [51]. Robust standard errors for the regression coefficients were calculated by using the *lmtest* and *sandwich* packages. Bootstrapping for risk differences was performed with the *boot* package.

## 3. Results

### 3.1. Sample Characteristics

From the 5256 INGER questionnaires that were sent to KORA participants, we received 3742 filled-in and valid questionnaires (a response rate of 71.2%). As model-based trees cannot deal with missing values, we had to exclude all of the participants with missing values in the outcome and exposure measures, as well as in the 40 sex/gender- and intersectionality-related covariates. This left a complete case sample of N = 2534 (67.7% of the whole sample) for our main analyses.

The sample characteristics of the complete case sample (N = 2534) can be seen in Table 1. The mean age of the participants was 62.7 years, explaining why only 47.6% of participants were employed, as many of the older participants were already retired.

For the distributions of the self-rated health outcome measure and the green space exposure measures in the complete case sample, see Table 2. The distributions of all 40 sex/gender- and intersectionality-related covariates in the complete case sample are presented in Appendix A.

A comparison of the complete case sample (N = 2534) with the whole sample (N = 3742) can be seen in Appendix A. Although very slight differences can be observed, e.g., in the percentage of participants with basic school education (4.5% less in the complete case sample), with a bad or moderate self-rated financial situation (3.6% less in the complete case sample), or with a bad self-rated health (2.3% less in the complete case sample), no essential distortions, which could bias the overall results, were induced by the exclusion of the participants with missing values.

### 3.2. Balance Diagnostics of the Propensity Score Adjustment

We used propensity scores as stabilized inverse probability weights in our analyses. It is therefore important to report the balance diagnostics of the propensity score adjustment. To assess the balance of the weighted dataset for the two subjective exposure measures we compared the absolute standardized mean difference before and after propensity score adjustment. Figure 2 shows the results for the *self-rated access to high-quality public green space* exposure measure. Before the adjustment the absolute standardized mean difference was quite high for several of the confounders, indicating imbalanced data. After adjustment all of the absolute standardized mean differences were below 0.1, indicating negligible imbalance. Similar results were found for the subjective exposure measure *self-rated greenness in the residential environment* (see Appendix A, p. 6, for the balance diagnostic plot).

For the two objective continuous exposure measures we compared the absolute exposure–covariate correlations before and after adjustment. The respective balance diagnostic plots can be seen in Appendix A, p. 8 (300 m buffer) and p. 10 (1000 m buffer). In both cases the dataset was more balanced with respect to the confounding covariates after adjustment. Overall, we can therefore assume that our data contain no major imbalances when using stabilized inverse probability weights in our model-based tree analyses.

Additionally, we detected neither visual nor numerical substantial violations of the positivity assumption for any of the exposures (see Appendix A, p. 2 and p. 5 for distributions of propensity scores amongst the exposed and non-exposed, and p. 12 for descriptive statistics of the stabilized inverse probability weights).

### 3.3. Association of Green Space with Self-Rated Health in the Complete Case Sample

For both subjective exposure measures we found a positive association with self-rated health within the complete case sample (see Table 3).

The raw odds ratio (OR) for the association between the *self-rated access to high-quality public green spaces* and self-rated health was 1.78 [95% confidence interval (CI): 1.44, 2.13]. In particular, 27.0% of the participants with access to lower quality public green spaces or no access to green spaces fell into the category bad self-rated health, in contrast to only 17.2% of the participants with access to high-quality public green spaces, as shown in Table 3. After propensity score adjustment, the OR was smaller but still significantly different from 1 (OR: 1.38 [95% CI 1.10, 1.73]). For the association between *self-rated greenness in the residential environment* and self-rated health, the raw OR was 1.65 [95% CI: 1.34, 2.04]. Table 3 shows that 31.8% of the participants with a less green residential environment fell into the category bad self-rated health, while among the participants with a very green self-rated residential environment only 22.0% fell into the category bad self-rated health. After propensity score adjustment, the OR was 1.39 [95% CI: 1.08, 1.80].

However, there was no association between the two objective green space measures based on NDVI data and self-rated health. For the greenness NDVI measure within a 300 m buffer around the residential address, the OR was 1.08 [95% CI: 0.97, 1.20] before propensity score adjustment and 0.97 [95% CI: 0.84, 1.13] after adjustment. For the greenness NDVI measure with a 1000 m buffer the OR was 1.08 [95% CI: 0.98, 1.19] before and 1.02 [95% CI: 0.89, 1.17] after propensity score adjustment. Note that the range of the NDVI values lies between 0.2 and 0.8; therefore, the OR and corresponding CI were rescaled to resemble changes of an increase of 0.1 of the NDVI value. The absence of an association of the NDVI measurements with self-rated health was also reflected by the close similarity of distributions of NDVI data stratified by participants with good and bad health (see Appendix A, p. 8 (300 m buffer) and p. 10 (1000 m buffer)).

### 3.4. Influence of Sex/Gender- and Intersectionality-Related Covariates on the Association of Green Space and Self-Rated Health

We used model-based trees to investigate whether the relationship of green space exposure with self-rated health was modified by any of our 40 sex/gender- and intersectionality-related covariates (see Appendix A for a list of all of the included covariates). Importantly, for all four exposure measurements, none of the seventeen covariates operationalizing the individual sex/gender self-concept were identified by the model-based tree analyses as splitting covariates. All of the identified subgroups were defined by intersections of covariates that were added to the analyses to operationalize the structural sex/gender relations (e.g., experience of discrimination) and intersectionality-related social categories.

#### 3.4.1. Self-Rated Access to High-Quality Public Green Space

Figure 3 shows the model-based tree for the exposure measure *self-rated access to high-quality public green space*. The first split was induced on the covariate *DiscriminationDisability*. All of the participants who strongly disagreed that they were disadvantaged because of a physical impairment were sent to the right branch of the tree, while everyone else was sent to the left branch. Within the left branch, the participants were further split based on the same covariate, *DiscriminationDisability*; the participants who rather disagreed were sent to terminal node 4, while all of the others were sent to terminal node 3. Within the right branch of the tree, the participants were further split based on the covariate *SGRelationsIncome*. The participants with a bad or moderate self-rated financial situation were sent to terminal node 6, while the participants with a good or very good financial situation were subdivided based on the covariate *SGRelationsSchoolEducation*. Terminal node 8 consisted of those participants with a basic secondary school degree, while participants with O-levels or A-levels were contained in terminal node 9.

In model-based trees the final subgroups have to be further investigated in order to determine whether the odds ratios differed between the two groups, indicating effect heterogeneity, or whether differences in the prevalence of good self-rated health were the core reason for splitting. A closer look at the plots for each subgroup (Figure 3) and the parameter estimates (see Appendix A, pp. 3–4) revealed that the splits based on the covariates *DiscriminationDisability* and *SGRelationsIncome* were predominantly determined by differing prevalences of good self-rated health in the resulting subgroups. The split based on the covariate *SGRelationsSchoolEducation*, however, provided evidence for effect heterogeneity between the two subgroups in node 8 and node 9. Both of these subgroups were characterized by no self-assessed discrimination based on disability, a good or very good self-rated financial situation, and exhibited a high prevalence of good self-rated health compared to the other three subgroups; however, while there was a strong positive association between the self-rated access to high-quality public green spaces and self-rated health in subgroup 8 (OR: 2.66 [95% CI: 1.44, 4.91]), there was no such green space–health relationship in node 9 (OR: 0.91 [95% CI: 0.60, 1.39]). This effect heterogeneity between the two subgroups was also found on the risk difference scale: while the risk difference (RD) between the participants exposed to high-quality public green spaces and the participants exposed to lower or no public green spaces was 0.11 [CI: 0.06, 0.16] in node 8, the RD in node 9 was −0.01 [CI: −0.04, 0.02]. The plots show that for the participants with high education levels in node 9 the prevalence of good health was high, regardless of access to high-quality green spaces; however, for the participants with a lower education level in node 8 the prevalence of good health was lower for the participants with no access to high-quality green spaces. Thus, in this context the participants with a lower education level might have benefited from access to high-quality green spaces, while the participants with a higher education level had a high prevalence of good health in any case.

#### 3.4.2. Self-Rated Greenness in the Residential Environment

The model-based tree for the exposure measure *self-rated greenness in the residential environment* can be seen in Appendix A, p. 6. The first three splits were identical to the tree described above for the exposure measure *self-rated access to high-quality public green space*. Thus, the participants were first split based on the covariate *DiscriminationDisability*, and then again based on *DiscriminationDisability* in the left branch and based on *SGRelationsIncome* in the right. Only the following splits in the right branch of the tree differed from the tree in Figure 3: The participants with a bad or moderate self-rated financial situation were split again based on the covariate *SGRelationsVocationalEducation*. The participants with a good or very good self-rated financial situation were further subdivided based on the covariate *DiscriminationSocialPosition*, with the participants neither agreeing nor disagreeing, rather agreeing, or strongly agreeing that they are disadvantaged based on their social position being sent to node 10, with the participants rather disagreeing or strongly disagreeing being sent to node 11. As before, a closer look revealed that splits based on the covariates *DiscriminationDisability* and *SGRelationsIncome* mainly affected differences in the prevalence of good self-rated health. Effect heterogeneity could be seen between the two subgroups in node 10 and node 11. As both of these subgroups were characterized by no discrimination based on disability and a good or very good self-rated financial situation, they showed a higher prevalence of good self-rated health compared to the other four subgroups. While there was a strong positive association between self-rated greenness in the residential environment and self-rated health in subgroup 10 (OR: 4.90 [95% CI: 1.42, 16.98], RD: 0.31 [CI: 0.08, 0.51]), there was no significant greenness–health relationship in node 11 (OR: 1.13 [95% CI: 0.67, 1.92], RD: 0.01 [CI: −0.03, 0.07]). The participants who felt disadvantaged because of their social position (despite having a good or very good self-rated financial situation) might have benefited from a very green residential environment, while the participants who felt less disadvantaged because of their social position had a high prevalence of good self-rated health no matter how green their residential environment was.

#### 3.4.3. Greenness within a 1000 m Buffer around the Residential Address

Even though we did not find any significant impact of the two objective greenness measures on self-rated health in the complete case sample, it may be the case that this relationship exists in certain subgroups. We therefore also performed model-based tree analyses for these two objective exposure measures.

Figure 4 displays the model-based tree for the NDVI measure within a 1000 m buffer. Splits based on the covariates *DiscriminationDisability*, *SGRelationsIncome* (two splits), and *DiscriminationWeight* led to five subgroups. A closer look at the plots for each subgroup (Figure 4) and the parameter estimates (see Appendix A, p. 11) revealed that all of the splits lead to subgroups with substantial differences in the prevalence of good self-rated health. The prevalence of good self-rated health was lowest in terminal node 3, a subgroup that feels disadvantaged based on disability (at least slightly—did not strongly disagree) and rated their financial situation as bad or moderate. In contrast, the highest prevalence was found in terminal node 9, a subgroup that strongly disagreed that they were disadvantaged based on disability or weight and rated their financial situation as good or very good; however, there was no significant objective greenness–self-rated health association in any of the subgroups, and there was no evidence for effect heterogeneity in this analysis, corroborated by the estimated effect estimates (see Appendix A, p. 11).

#### 3.4.4. Greenness within a 300 m Buffer around the Residential Address

The model-based tree for the NDVI measure with a 300 m buffer can be seen in Appendix A, p. 9. Splits based on the covariates *DiscriminationDisability* (two splits), *SGRelationsIncome*, *DiscriminationWeight*, and *DiscriminationAge* constituted six subgroups. The findings were consistent with the 1000 m buffer analysis. Similarly, plots and parameter estimates (see Appendix A, p. 9) uncovered the fact that most splits resembled differences in the prevalence of good self-rated health. The only exception was the split induced by *DiscriminationAge*, which led to a significant negative objective greenness–self-rated health association in terminal node 5 in comparison to node 6, with no significant relationship; however, an unusual peak for low NDVI values in the conditional density plot of the rather small subgroup contained in node 5 (n = 101) indicated that parameter estimates may be biased. Thus, this result has to be interpreted with caution.

### 3.5. Additional Analysis for the INGER KORA FIT Subsample with 53 Sex/Gender- and Intersectionality-Related Covariates

From the whole INGER sample, 70.1% of the participants (N = 2624) took part in the KORA FIT survey, which was conducted shortly before the INGER survey, i.e., 2018/2019. For this subsample of the participants, 13 additional covariates from the KORA FIT survey were available (see Appendix A). After excluding all of the participants with missing values in the outcome measures, exposure measures, and 53 sex/gender- as well as intersectionality-related covariates, N = 1687 participants (64.3% of the whole INGER KORA FIT sample) were used for these additional analyses.

As was the case for the main analysis, this subsample showed positive associations between the two subjective green space measures and self-rated health, but no associations between the two objective green space measures based on NDVI data and self-rated health (see Appendix A, p. 1 (*self-rated access to high-quality public green space*), p. 5 (*self-rated greenness in the residential environment*), p. 8 (*NDVI—300 m buffer*), and p. 10 (*NDVI—1000 m buffer*)).

As can be seen in the balance diagnostic plots in Appendix A, p. 2 (*self-rated access to high-quality public green space*), p. 6 (*self-rated greenness in the residential environment*), p. 8 (*NDVI—300 m buffer*), and p. 10 (*NDVI—1000 m buffer*), the dataset was more balanced with respect to the confounding covariates after propensity score adjustment than before for all four exposure measures. The positivity assumption was also not violated (see Appendix A p. 2 and p. 5 for distributions of propensity scores amongst the exposed and non-exposed and p. 12 for descriptive statistics of the stabilized inverse probability weights).

The model-based trees for all four exposure measures (see Appendix A, p. 3 (*self-rated access to high-quality public green space*), p. 6 (*self-rated greenness in the residential environment*), p. 9 (*NDVI—300 m buffer*), and p. 11 (*NDVI—1000 m buffer*)) again did not show effect heterogeneity based on any of the 17 covariates operationalizing the individual sex/gender self-concept; however, they included several splits based on some of the additionally added covariates operationalizing intersectionality-related social categories and psychosocial factors.

For instance, in all four trees the first split was induced by the additional covariate *SGRelationsMobility*, separating participants depending on their ability to move around. Additionally, covariates representing psychosocial factors were often identified as splitting variables: the covariate *PerceivedStress* induced splits in both the trees for the subjective exposure measures and in the tree for the NDVI measure with a 300 m buffer, while the covariates *OptimismPessimism* and *LifeSatisfaction* led to splits in the tree for the exposure measure *self-rated greenness in the residential environment*. Despite this appearance of additional variables, the two covariates that were most often identified in the main analyses still led to splits in these additional analyses: *DiscriminationDisability* induced splits in both the trees for the subjective exposure measures and in the tree for the NDVI measure with a 1000 m buffer, while *SGRelationsIncome* led to splits in both trees for the NDVI exposure measures.

As was the case in the main analyses, a closer look at the plots for the subgroups and the corresponding parameter estimates (see Appendix A, pp. 3–4 (*self-rated access to high-quality public green space*), p. 7 (*self-rated greenness in the residential environment*), p. 9 (*NDVI—300 m buffer*), and p. 11 (*NDVI—1000 m buffer*)) revealed that most splits were predominantly determined by differing prevalences of good self-rated health in the subgroups.

Overall, three splits seemed to induce effect heterogeneity: First, in the tree for the exposure measure *self-rated greenness in the residential environment*, the split based on the covariate *LifeSatisfaction* led to terminal nodes 7 and 8 with differential effects (see Appendix A, pp. 6–7). While both subgroups consisted of participants who had no problems moving but had rather high levels of perceived stress, the participants in node 7 were less satisfied with their lives right now compared to the participants in node 8. The participants in node 8 had an overall higher prevalence of good self-rated health, but showed no impact of greenness on self-rated health, while the participants in node 7 showed a lower overall prevalence of good self-rated health but might have benefitted from a very green residential environment.

Second, in the tree for the NDVI measure with a 300 m buffer, a split based on *SGRelationsMobility* led to effect heterogeneity between terminal nodes 8 and 9 (see Appendix A, p. 9). The participants in node 9 had moderate or big problems moving and, as was the case for the overall sample in this analysis, showed no association between the NDVI greenness measure and self-rated health. The participants in node 8, on the other hand, had little problems moving and might have benefitted from greener environments within a 300 m buffer around their residential addresses.

Third, in the tree for the NDVI measure with a 1000 m buffer, the split based on *SGRelationsIncome* induced effect heterogeneity between terminal node 3 and intermediate node 4 (see Appendix A, p. 11). The participants with no problems to move and a good or very good self-rated financial situation in node 4 showed no association between the NDVI greenness measure and self-rated health, as was the case for the overall sample in this analysis and the two terminal nodes, 5 and 6, resulting from node 4. In comparison, the participants in terminal node 3 with no problems moving but a bad or moderate self-rated financial situation showed a positive greenness health association.

### 3.6. Additional Analysis with a Binary Sex/Gender Variable

The women and men (defined by the variable *SexAtBirth*) in our sample did not differ significantly with respect to the outcome of self-rated health (women: 74.8% good self-rated health, men: 77.4% good self-rated health; χ^2^ = 2.2859, *p*-value = 0.1306). There were also no significant differences between the two groups for any of the four green space exposure measures, i.e., *self-rated access to high-quality public green spaces* (women: 31.2% access to high-quality green spaces, men: 29.8% access to high-quality green spaces; χ^2^ = 0.54033, *p*-value = 0.4623), *self-rated greenness in the residential environment* (women: 79.6% very green environment, men: 78.8% very green environment; χ^2^ = 0.19027, *p*-value = 0.6627), NDVI greenness within a 300 m buffer (women: mean = 0.47, SD = 0.09, men: mean = 0.47, SD = 0.08, t = 1.1396, *p*-value = 0.2546), and NDVI greenness within a 1000 m buffer (women: mean = 0.50, SD = 0.09, men: mean = 0.51, SD = 0.09, t = 1.7973, *p*-value = 0.0724).

In order to directly compare our results with former studies that use a binary sex/gender variable, we tested whether the binary sex/gender variable *SexAtBirth* modified the green space–health association. This was assessed by a likelihood ratio test that compared the models with and without an exposure × *SexAtBirth* interaction term. For the two subjective exposure measures the results were mixed. The positive association between the *self-rated access to high-quality public green spaces* and self-rated health was modified by the binary variable *SexAtBirth* (χ^2^ = 7.99, *p* = 0.0047), with men showing a positive association (OR: 1.99 [95% CI: 1.43, 2.80]) while women did not (OR: 1.07 [95% CI: 0.83, 1.41]); however, we found no modification by the binary sex/gender variable for the positive *self-rated greenness in the residential environment* and self-rated health association (χ^2^ = 0.80, *p* = 0.372).

For both objective exposure measures the greenness × *SexAtBirth* interaction term was significant (NDVI 300 m: χ^2^ = 5.02, *p* = 0.025; NDVI 1000 m: χ^2^ = 8.24, *p* = 0.0041), with a tendency toward an inverse association; however, for the greenness within a 300 m buffer NDVI measure neither men (OR: 1.13 [95% CI: 0.95, 1.33]) nor women (OR: 0.87 [95% CI: 0.75, 1.01]) showed a significant greenness–health association. Yet, for the greenness within a 1000 m buffer NDVI measure, men showed a positive greenness–health association (OR: 1.21 [95% CI: 1.04, 1.41]), while for women this association was not significant (OR: 0.89 [95% CI: 0.78, 1.02]).

## 4. Discussion

In this study, we investigated whether sex/gender had an impact on the association between green space, measured both subjectively and objectively, and self-rated health. To comprehensively assess the relevance of sex/gender, we defined it as a multidimensional concept, operationalized by multiple sex/gender- and intersectionality-related covariates [20]. None of the sex/gender covariates representing the individual sex/gender self-concept (i.e., the dimensions *sex assigned at birth, current sex/gender identity, internalized sex/gender roles*, and *externalized sex/gender expressions*, see Bolte [20]) led to subgroups with heterogeneous green space–self-rated health effects in our model-based tree analyses. We did, however, detect effect heterogeneity based on intersections of covariates that contribute to explaining structural sex/gender relations (e.g., experience of discrimination) and intersectionality-related social categories. Amongst participants with no self-assessed discrimination based on disability and a good or very good self-rated financial situation, those with a lower education level or a feeling of discrimination based on social position showed a positive green space-self-rated health association, while participants with a higher education level or no feeling of discrimination based on social position had a high level of good self-rated health regardless of the availability of green spaces.

### 4.1. Association of Green Space with Self-Rated Health in the Complete Sample

In general, exposure to green space is considered to have a positive impact on health [1,2,3,4]; however, a closer look at the literature shows that this positive association is not always consistent [52,53,54] but depends on many factors, such as the measured health outcomes [54,55] or the green space exposure measure [35]. In our study, the presence of a positive green space–self-rated health association in the complete sample was influenced by the operationalization of green space. While we found a positive effect for both of our subjective green space measures, we did not see this relationship between the greenness captured by NDVI data and self-rated health in the complete sample. First of all, this underlines the importance of including both subjective and objective exposure measures in green space research studies, as these do not always coincide [35,36]. Our results might suggest that the positive association of green space with self-rated health is mostly influenced by the subjective feeling of living in a very green environment, rather than objectively measured greenness. For instance, in areas with many private gardens behind walls, objective greenness measures cannot capture whether these areas are accessible to all residents; however, the difference between subjective and objective measurements is likely to depend on several other factors. For example, we might have found different results for other health outcomes, and, in fact, results differed between distinct subgroups, as our model-based tree analyses showed.

### 4.2. Influence of Sex/Gender- and Intersectionality-Related Covariates on the Green Space–Self-Rated Health Relationship

Our model-based trees did not indicate any relevance of the covariates representing the individual sex/gender self-concept [20] for the green space–self-rated health relationship; however, in the different trees effect heterogeneity was observed between subgroups defined by covariates that contribute to explaining structural sex/gender relationships and intersectionality-related social categories. In all cases we saw a similar picture: the more disadvantaged subgroups (e.g., the less educated, the more discriminated against based on social position) showed a positive association between green space and self-rated health; thus, these groups might have benefitted from green spaces in their residential environments. In contrary, the more advantaged subgroups did not show any association between green space and self-rated health, but had a higher prevalence of good self-rated health independent of green space. These results are in line with a recent review that showed that green space had a greater protective effect for people with a lower SES than for more affluent people [7]. As the authors of this review suggested, more disadvantaged groups may depend more on proximate green space as they have less access to other health-promoting resources [7].

A direct comparison with former studies that examine effect modification through sex/gender of the green space–health association can only be carried out by using a binary sex/gender variable, as none of the former studies considered the complexity of sex/gender adequately [12,20]. Our results, when using only a binary sex/gender variable, suggested that men have a stronger positive green space–self-rated health relationship than women for two of the four exposure measures (*self-rated access to high-quality public green spaces* and NDVI greenness within a 1000 m buffer). Sillman [12] suggested the opposite in their recent review, suggesting that the green space–physical health association was stronger for women than for men; however, they also showed that this was mostly the case for obesity-related health outcomes as well as mortality, and their results for the outcome measure general health were mixed, with the majority of studies showing no difference between men and women, while some showed stronger effects for men and others for women. Thus, our results, when using a binary sex/gender variable, can be added to this heterogeneity of findings as we detected no difference between men and women for two of our green space exposure measures and stronger effects for men for the other two exposure measures. Yet, our results can also not be explained by Richardson and Mitchell’s [17] suggestion that differences found between studies concerning the effect modification by sex/gender may be explained by differences in exposure assessment. They proposed that women may be more influenced by the quality of green space, while for men the effect can be demonstrated by objective green space quantity measures [17]. We could not find results supporting this hypothesis, despite the fact that we explicitly compared both a measure of the quality of green space and an objective green space measure. In fact, our measure concerning the quality of public green space showed a stronger green space quality–health association for men. This heterogeneity of results may underline the fact that solely using a binary sex/gender covariate cannot capture the complexity of the underlying situation.

Our results based on the comprehensive operationalization of a complex multidimensional sex/gender concept suggested that covariates that contribute to explaining structural sex/gender relationships and intersectionality-related covariates were more relevant for identifying subgroups that benefit the most from green spaces than covariates operationalizing the individual sex/gender self-concept. Thus, structural aspects referring to social disadvantages and the gendered distribution of societal power seemed to be relevant for environmental health promotion. This is an important result for urban planning, as providing disadvantaged groups with better access to public green spaces has the potential to increase health equity [1,7].

### 4.3. Analytic Intersectional Approach against the Background of the Current Scientific Debate

In recent years the scientific debate on appropriate methods for quantitative intersectional research has become more intensive [31,56,57]. We consider sex/gender from an intersectional perspective and follow an intercategorical intersectional approach [58] to explore differences between different intersections of sex/gender and further social categories. Decision trees have been identified as promising methods when interested in comparing large numbers of intersections [56]; however, decision tree methods, such as CARTs [28] and CITs [29], which have so far been used [59,60] and compared [57] in quantitative intersectional research, can only be used for descriptive intersectional research [31]. In this study, however, we wanted to explore whether we would be able to identify intersections with differential effects of an environmental exposure on a health outcome, and thus followed a more analytic intersectional approach as defined by Bauer and Scheim [31]. We therefore used model-based trees, as these do not only find subgroups with heterogeneous health outcomes but can simultaneously determine subgroups with differential exposure–outcome associations.

It is also important to remember that intersectionality is a framework and that not all of the approaches within this framework are directly comparable with each other. This is best demonstrated by comparing our analysis strategy and results with a recent study that also used an intersectional approach to examine the associations between urban green space and self-perceived health [61]. In their study, Rodriguez-Loureiro [61] considered pre-defined intersections between the three social categories gender, education, and migrant background. They found that, for women, the strongest beneficial associations were found in the less educated, regardless of migrant background, while for men those with the highest education levels and of Belgian origin showed the strongest association. These results are not in line with our results. Although both we and Rodriguez-Loureiro [61] referred to an intersectionality framework, the methodological approaches are far from similar. They considered three social categories, including a binary variable of sex/gender. We, on the other hand, considered sex/gender to be multidimensional [20] according to current gender theoretical concepts and therefore operationalized sex/gender by using 17 covariates representing the individual sex/gender self-concept and 23 covariates operationalizing structural sex/gender relationships, including further intersectional categories such as education. Rodriguez-Loureiro [61] examined and compared all possible intersections based on gender, education, and migrant background. This approach was, however, not feasible and reasonable when studying sex/gender more comprehensively by considering 40 covariates, as this would have resulted in a very high number of possible intersections. We, therefore, used an explorative, data-driven approach with model-based trees. We wanted to identify the most differential subgroups with respect to either the prevalence of good self-rated health and/or the effect of green space exposure on self-rated health. Using an explorative approach enabled us to identify intersections previously not thought of. We found, for example, that a combination of the covariates discrimination based on disability, self-rated financial situation, and school education led to two subgroups, i.e., intersections, with differential green space quality–health effects. Nevertheless, as is always the case for exploratory research, conclusions should be tested in an independent sample in a next step.

### 4.4. Strengths and Limitations

As it is always the case for observational, cross-sectional studies, causal conclusions about the examined exposure–outcome relationships cannot be drawn. Although we employed inverse probability weighting based on propensity scores to adjust for confounding variables and created a pseudo-randomized sample, we have to recall that this method has its limitations, e.g., the possibility of unmeasured confounders.

We applied model-based recursive partitioning to detect subgroups exhibiting different exposure–outcome associations. A major advantage of this method over classical approaches is that it can handle a large number of covariates. While regression models with interaction terms depend on a-priori knowledge of which interactions to include, this data-driven approach identifies suitable covariates without further model specifications. Moreover, complex interactions between several covariates can be uncovered, which is crucial for intersectional research. While there are various methods with which to detect differences in exposure–outcome relationships for binary exposures, there exists only a limited number of tools for handling continuous exposures, such as NDVI measurements. Model-based recursive partitioning provides a unified framework for analyzing these relationships, allowing for all types of exposures and outcomes. Furthermore, the possibility to include inverse probability weights allowed us to adjust for confounders. Finally, the resulting decision trees and their visualization allowed easy interpretation and description of the detected subgroups.

A few drawbacks of this modeling approach need to be addressed: First, most decision tree algorithms provide the use of surrogate splits to handle missing data in any of the splitting variables [28]; however, there exists no such implementation for model-based recursive partitioning. Thus, we needed to exclude about 32% of our data from the analyses. Although imputation is feasible, we decided not to pursue this approach since it could have introduced additional bias. Second, the algorithm fits a logistic regression model to each node, resulting in estimates of an intercept and an exposure coefficient. Thus, splits identified by the structural change test may be due to changes in the intercept and therefore do not grant insight into whether effect heterogeneity on the odds ratio scale was present or not; however, since this approach is parametric, goodness of fit is important for reducing bias. Thus, the resulting subgroups locally described the data well. Whether subgroups differed only by the prevalence of good self-rated health and/or exposure effect had to be investigated in a second step. Large differences in the prevalence of good self-rated health between certain social groups might mask not very pronounced effect heterogeneity, especially if the maximum depth of the tree is limited to ensure the interpretability of the results.

A major strength of our study is the comprehensive assessment of sex/gender, with at least 40 covariates based on a theoretical sex/gender concept. A further strength is the comprehensive operationalization of green space. We used both subjective and objective measurements of exposure to green space and also considered the quality of public green spaces. Yet, some exposure misclassification might have occurred for participants in rural areas, as they might have interpreted privately owned forests or meadows as green spaces not publicly available to them, negating having access to public green spaces, although these privately owned green spaces might still be accessible to a certain extent and therefore beneficial to their health. The KORA study population is well-characterized, and we achieved a high response rate in our KORA INGER survey; however, the study population is rather homogeneous in terms of age, ethnicity, social disparities, and availability of public green spaces when compared to metropolitan populations. The application of our approach to a more heterogeneous urban population would therefore be desirable. Future research might also benefit from exploring further quality aspects of green spaces (e.g., safety issues) and considering the actual usage of public green spaces as opposed to only their availability.

## 5. Conclusions

A theory-based, comprehensive assessment of the relevance of sex/gender for the association of green space and self-rated health showed effect heterogeneity based on covariates representing structural aspects from an intersectionality perspective, although these covariates did not show intersectional structuring of sex/gender dimensions. Amongst participants with no self-assessed discrimination based on disability and a good or very good self-rated financial situation, those with a lower education level or feeling of discrimination based on social position showed a positive green space–self-rated health association, while participants with a higher education level or no feeling of discrimination based on social position had a high level of good self-rated health regardless of the availability of green spaces. In general, there were differences between the subjective and objective exposure measures, with a positive association for both subjective green space measures and self-rated health, but no relationship between greenness captured by NDVI data and self-rated health. This underlines the importance of reporting different types of green space measures in future studies. Methodological approaches, such as model-based recursive partitioning, should be further employed to detect subgroups that exhibit different exposure–outcome associations, with the possibility of integrating multiple sex/gender- and intersectionality-related covariates as potential effect modifiers.

## Figures and Tables

**Figure 1 ijerph-20-05241-f001:**
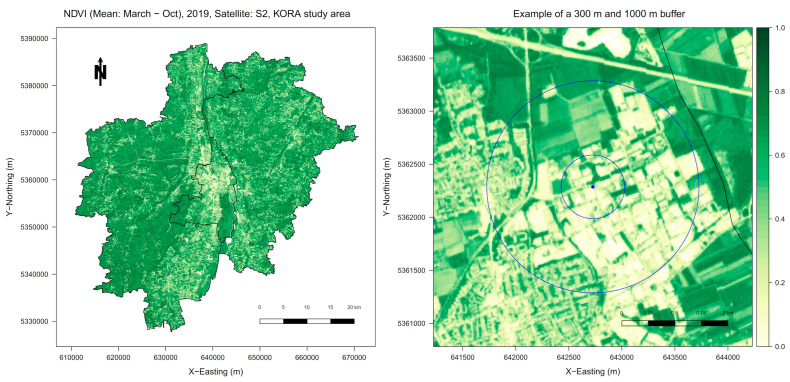
Visualization of the NDVI (normalized difference vegetation index) data. The **left** panel shows the NDVI data recorded by the Sentinel-2 satellite with a ground resolution of 10 m, collected between March and October 2019 for the KORA (Cooperative Health Research in the Region of Augsburg) study area, including the city Augsburg (in the middle of the map) and two adjacent districts in Southern Germany. The **right** panel demonstrates the applied 300 m and 1000 m buffers around an exemplary residential address. © GeoBasis-DE/Bundesamt für Kartographie und Geodäsie (BKG, 2019).

**Figure 2 ijerph-20-05241-f002:**
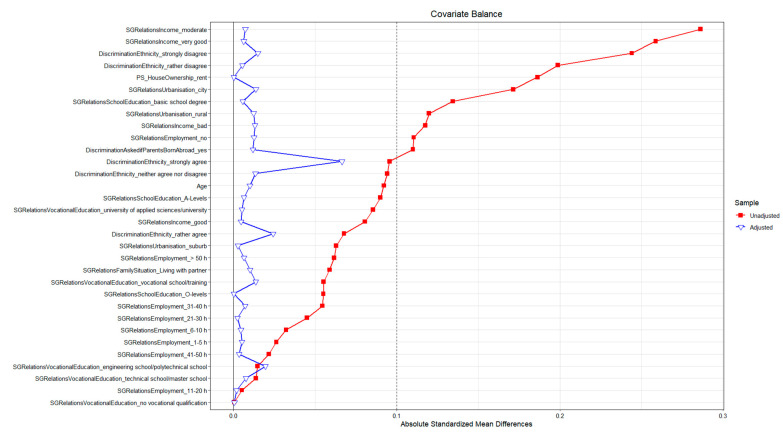
Balance diagnostics for the *self-rated access to high-quality public green space* exposure measure, showing absolute standardized mean differences before (unadjusted—red line with squares) and after (adjusted—blue line with triangles) weighting. Differences in the adjusted sample were below 0.1 (dotted vertical line), indicating good covariate balance.

**Figure 3 ijerph-20-05241-f003:**
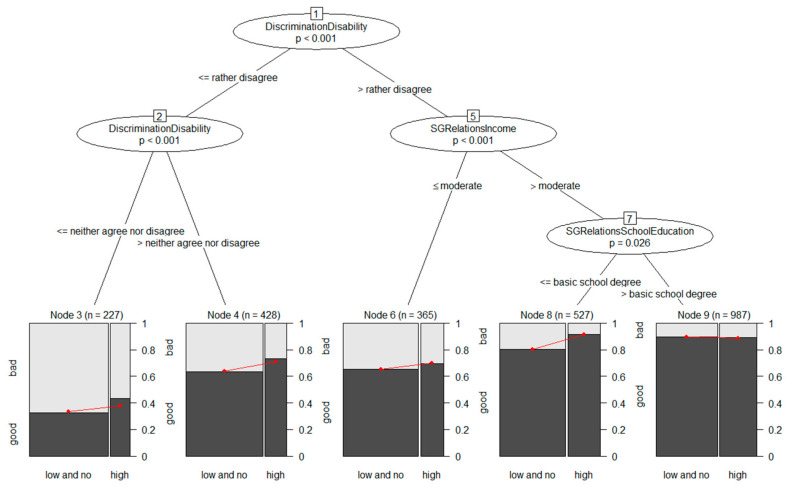
Generalized linear model tree for the exposure *self-rated access to high-quality public green space*. Each terminal node in the bottom row contains a plot of the relationship between exposure and self-rated health. Red lines correspond to parameter estimates (intercept and exposure coefficient) obtained by weighted logistic regressions.

**Figure 4 ijerph-20-05241-f004:**
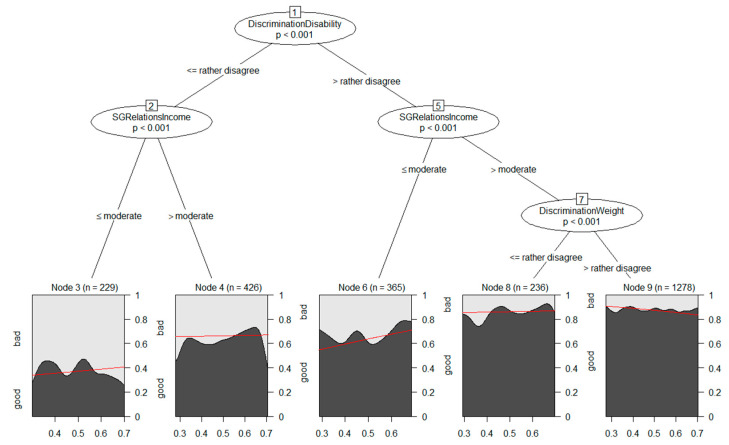
Generalized linear model tree for the NDVI exposure measure greenness within a 1000 m buffer around the residential address. Each terminal node in the bottom row contains a plot of the relationship between exposure and self-rated health. Red curves depict the estimated conditional probability curves obtained by weighted logistic regressions.

**Table 1 ijerph-20-05241-t001:** Sample characteristics in the complete case sample.

Question in INGER KORA Survey ^1^	Response Categories and Distribution in the Complete Case Sample; N = 2534 (100%)	
Age distribution	Continuous	(years)
	62.7	= Mean
	8.9	= SD
	43.0	= Min.
	90.0	= Max.
What is your current sex/gender identity? (Multiple responses possible.)	1334 (52.6)	= Female
	1184 (46.7)	= Male
	2 (0.1)	= Transsexual
	0 (0.0)	= Another identity
	8 (0.3)	= I don’t want to classify
	2 (0.1)	= Female and I don’t want to classify
	2 (0.1)	= Male and I don’t want to classify
	2 (0.1)	= Male and another identity
What sex were you assigned at birth?	1339 (52.8)	= Female
	1195 (47.2)	= Male
	0 (0.0)	= Divers/intersexual
What is your highest school education?	1062 (41.9)	= Degree after German basic secondary school (Hauptschulabschluss)
	793 (31.3)	= German O-Levels (Mittlere Reife)
	679 (26.8)	= German A-Levels (Abitur)
What is your highest vocational qualification?	157 (6.2)	= No vocational qualification
	1358 (53.6)	= Vocational school/apprenticeship
	504 (19.9)	= Technical school/master school
	22 (0.9)	= Engineering school/polytechnical school
	493 (19.5)	= University of applied sciences/university
Are you employed?	1206 (47.6)	= Yes
	1328 (52.4)	= No
How do you assess your financial situation?	295 (11.6)	= Very good
	1645 (64.9)	= Good
	559 (22.1)	= Moderate
	35 (1.4)	= Bad
Do you live…?	2123 (83.8)	= In your own property
	411 (16.2)	= For rent
Distribution of participants by degree of urbanisation.	902 (35.6)	= City
	1091 (43.1)	= Suburb
	541 (21.3)	= Rural

^1^ The questions were originally asked in German.

**Table 2 ijerph-20-05241-t002:** Distributions of the outcome measure and the four exposure measures in the complete case sample.

Subjective Assessment within the INGER KORA Survey ^1^	Response Categories and Distribution in the Complete Case Sample; N = 2534 (100%)	
**Self-Rated Health ^2^** How is your health in general?	269 (10.6)	= Very good
	1657 (65.4)	= Good
	550 (21.7)	= Moderate
	57 (2.2)	= Bad
	1 (0.04)	= Very bad
**Self-Rated Access to High-Quality Public Green Spaces** Based on a combination of three questions: Are there publicly accessible green spaces (e.g., parks, forests, meadows) in your neighbourhood? The publicly accessible green spaces in my neighbourhood are well maintained. The publicly accessible green spaces in my neighbourhood are of high quality.		
	774 (30.5)	= High-quality green spaces
	1760 (69.5)	= Lower quality or no green spaces
**Self-Rated Greenness in the Residential Environment ^3^** How green is your neighbourhood? (From green strips along the road to gardens and parks.)	2008 (79.2)	= Very green
	477 (18.8)	= A little green
	44 (1.7)	= Hardly green
	5 (0.2)	= Not green at all
**Objective Exposure Measurement**	**Continuous Variables**	
**Greenness within a 300 m buffer around the residential address ^4^**	0.47	= Mean
	0.09	= SD
	0.16	= Min.
	0.73	= Max.
	0.47	= Median
	0.41	= Q1
	0.53	= Q3
**Greenness within a 1000 m buffer around the residential address ^4^**	0.50	= Mean
	0.09	= SD
	0.27	= Min.
	0.71	= Max.
	0.50	= Median
	0.44	= Q1
	0.57	= Q3

^1^ The questions were originally asked in German. ^2^ Note that for the analyses this variable was dichotomized: “very good” and “good” were grouped together as “good”, while “moderate”, “bad”, and “very bad” were grouped together as “bad”. ^3^ Note that for the analyses this variable was dichotomized: “a little green”, “hardly green”, and “not green at all” were grouped together as “less green”. ^4^ NDVI data, calculated from several satellite images between March and October 2019. Negative pixels of the NDVI map were excluded prior to assignment to the residential addresses.

**Table 3 ijerph-20-05241-t003:** Distribution of self-rated health by the two subjective exposure measurements.

Self-Rated Access to High-Quality Public Green Spaces	Good Self-Rated Health	Bad Self-Rated Health
High-quality green spaces *N* = 774 (100%)	641 (82.8)	133 (17.2)
Lower quality or no green spaces *N* = 1760 (100%)	1285 (73.0)	475 (27.0)
**Self-Rated Greenness in the Residential Environment**	**Good Self-Rated Health**	**Bad Self-Rated Health**
Very green *N* = 2008 (100%)	1567 (78.0)	441 (22.0)
Less green *N* = 526 (100%)	359 (68.2)	167 (31.8)

## Data Availability

The data that support the findings of this study are available from the KORA study team of the Helmholtz Zentrum München, but restrictions apply to the availability of these data, which were used under license for the current study and so are not publicly available.

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
