# Peer review of "Do Multiple Sex/Gender Dimensions Play a Role in the Association of Green Space and Self-Rated Health? Model-Based Recursive Partitioning Results from the KORA INGER Study"

_ijerph, 2023, doi:10.3390/ijerph20075241_

Round 1
Reviewer 1 Report
Dear Authors
I thoroughly enjoyed reading your manuscript. I believe it reads well, is highly relevant for the field, and presented in a well-structured manner.
This is a very timely and policy relevant topic. You have explored whether sex/gender, operationalized through multiple sex/gender and intersectionality-related covariates, modifies greenspace/self-rated health associations. The particular strengths of your study is the combination of subjective and objective greenspace measures and the comprehensive consideration of sex/gender as a multidimensional concept. None of the covariates operationalizing an individual sex/gender self-concept led to subgroups with heterogeneous effects.
Key findings: i) Those with a lower education or a feeling of discrimination based on social position showed a positive greenspace/self-rated health association, while participants with a higher education or no feeling of discrimination based on social position had a high level of good self-rated health regardless of the availability of green spaces. ii) There appear to be differences between perceived and actual greenspace access and quality, with an association between perceived access/quality and self-rated health both no association between actual access/quality and self-rated health.
I have added comments in the attached pdf while reading the manuscript. Initially I had a few concerns, but later on in the manuscript you have addressed them all. Perhaps you want to consider addressing the mentioned concerns earlier on in the paper. I also believe you have many figures and tables as supplementary material. Would it be possible to include, at least some of them, in the main paper? This would address some of my initial concerns and make it more accessible to the reader. I appreciate this might have to do with lack of space!
General comment:
Clear and concise introduction.
The applied methods are described very well, and so the results would be easy to reproduce.
Methods, in general, very well described and any changes made to the methodologies are justified well.
The data is interpreted appropriately and consistently and the analysis is explained well.
The conclusions are consistent with the results presented.
Line 190: Was the meaning of 'high quality' described further? 'Quality' can mean many things: opportunity for walking your dog, for playing football, for sitting, for socialising, for peace and quiet, increase biodiversity etc.
Line 202: Did you use images to illustrate these categories?
Section 2.6 and 2.7: I have not used these methods myself and thus cannot comment on the technical details. However, the language used is approachable and I am able to understand the broader context, without having used these techniques in my own research.
Line 419: This is the first time 'self-rated' residential environment is mentioned. It is vital for the validity of the results, that it is made clear, that it is self-rated quality and quantity of greenspace and not an attempt to assess quality and/or quantity according to any predetermined standard. If it is not clearly described, the readers will make their own assumptions, which I did.
Line 422: Does this mean perception of quality and/or quantity is more important than actual quality/quantity?
Line 482 and 512: This is very clear and concisely described
Line 565: A very interesting finding
Line 669: Very well summarised. You have turned a quite long a complicated story in to a succinct paragraph.
Line 684: I believe this is a very interesting finding
Line 702 and 742: Very interesting and succinct discussion
Line 793: Good and comprehensive discussion on strengths and limitations.
Line 839: Clear and concise conclusion. I believe the results on differences between perceived/actual greenspace access and quality (line 422) should be included in the conclusion as a key finding.
Overall, a very well written and interesting manuscript.
I wish you good luck with the publication of you manuscript.

Reviewer 2 Report
The study is interesting and valuable. And the article is complete and clearly structured. There are no major problems with the article. The authors clearly describe the relationship between gender-dimensions in green space and health.
1. This is an interesting study in which the authors explore the role of gender in green technology and self-rated health. This is a relatively rare study. It has some research value. 2, The data and research methods of the article study are reasonable. However, the research process is somewhat complicated, and some parts are more difficult to understand. For example, in the section 2.Materials and Methods, there are 10 vignettes. It is suggested to add a research structure diagram to present the research process in a diagrammatic way. 3. Some important research data are suggested to be shown in the form of attachments. For example, questionnaire, population, etc.
Accept in present form.
Reviewer 3 Report
This is a carefully structured paper.
However, the lack of an important perspective on the health benefits of green spaces has led to uninformative results in the paper. The research design should be reconsidered so as to add knowledge familiar with green spaces, forests, trees, and green trails.
Comment
Chapter 2: Focusing on public green spaces, but there are private and company-owned forests, farmlands and trees that are effective. Users cannot distinguish public or not.
Another question is about the entire green space, but it overlaps with the public. What does neighborhood mean?
Chapter 4: The result is not useful as an academic paper. A known result. There was no effective result for the target gender difference may be due to the design about green space.
advice
Gender differences have more to do with quality of green than with quantity of green. Add quality as well as quantity to NDVI. The health benefits of shrub forests differ greatly from those of aged forests. There is a difference in the stress reduction effect between a safe forest trail and a dark trail.
For example, parks with well-maintained walking-trails give women a sense of security and reduce stress, but unmaintained walking-trails are stressful. I think the effect of stress reduction is small for men.
In other words, you can compare the effect on residents who live near parks with safe walking paths in the city and on residents who live in unsafe (insufficiently managed) green spaces by gender.
Reviewer 4 Report
2.2. Study Population
- It would more beneficial for the readers if the authors can provide a location map of the study area (i.e., Augsburg and two adjacent counties) and include location or parks and green spaces in these study areas.
3.4.2. Greenness in the residential environment
- Please provide a map showing the level of "greenness" within the 300 and 1000 m buffers to provide a better overview to the readers.
Round 2
Reviewer 3 Report
Did not answer correctly due to lack of specialization in green spaces.
